# Radiation Angle Estimation and High-Precision Pedestrian Positioning by Tracking Change of Channel State Information[note 1]

**DOI:** 10.3390/s20051430

**Published:** 2020-03-05

**Authors:** Wataru Komamiya, Suhua Tang, Sadao Obana

**Affiliations:** Graduate School of Informatics and Engineering, The University of Electro-Communications, Tokyo 182-8585, Japan; w.komamiya@can.lab.uec.ac.jp (W.K.); sa-obana@uec.ac.jp (S.O.)

**Keywords:** angle estimation, pedestrian positioning, channel state information, doppler shift, pedestrian-to-vehicle communication

## Abstract

P2V (pedestrian-to-vehicle) communication, in which a pedestrian’s mobile device notifies its position to nearby vehicles in order to prevent pedestrian accidents, has attracted much research interest recently, but its performance largely depends on the precision of pedestrians’ positioning. Pedestrian positioning is generally performed by using GPS (Global Positioning System), and its precision may greatly degrade in urban canyons. To improve positioning precision of pedestrians, it was proposed that vehicles around pedestrians be used as anchors beside satellites. In this method, a pedestrian device overhears V2V (vehicle-to-vehicle) communication signals which carry the vehicle position and calculates a pedestrian’s position using pedestrian–vehicle distance/angle information estimated from CSI (channel state information) of the V2V signals. However, angle estimation typically depends on the number of antennas of the pedestrian device. In this paper, we propose a new method to estimate signal radiation angle by tracking temporal change of CSI caused by vehicle movement during signal transmission and investigate its application to precise pedestrian positioning. Three-dimensional ray tracing simulations confirm that compared to the base method using eight antennas, the proposed method with a single antenna reduces average angle error from 13 degrees to 3.9 degrees, and reduces average positioning error from 2.49 m to 0.78 m.

## 1. Introduction

V2V (vehicle-to-vehicle) and P2V (pedestrian-to-vehicle) communications have been studied extensively in order to prevent traffic accidents. In V2V communication, each vehicle periodically (e.g., 100 ms) broadcasts its position/speed information, which enables adjacent vehicles to predict potential collisions in advance. It has been put into practical use in Japan since 2015 using the 700 MHz band [1], and many new vehicles are equipped with this system [2,3]. In P2V communication, a pedestrian’s mobile device broadcasts its position information to nearby vehicles [4]. These systems can detect a vehicle/pedestrian in blind spots, which is otherwise impossible, by using a vehicle’s camera or LiDAR (light detection and ranging) sensor. Yet the performance of P2V communication heavily depends on the positioning precision of pedestrians.

Outdoor positioning is generally performed by using GNSS (Global Navigation Satellite System) such as GPS (Global Positioning System) and Galileo, with a satisfactory precision in open sky environments. Yet in urban canyons, satellite signals are often obstructed (the number of available satellites is not sufficient for positioning) or reflected by roadside buildings (error in pseudo-range estimation caused by reflected signal degrades positioning precision) [5]. Recently, there is a trend of integrating multiple satellite systems (e.g., GPS, Beidou, Galileo, GLONASS (Global Navigation Satellite System), and QZSS (Quasi-Zenith Satellite System), which helps to increase the number of available satellites; but in urban canyons, when available satellites are concentrated near the zenith, the performance improvement is limited due to the unbalanced distribution of satellites [6].

This has different impacts on pedestrians and vehicles. Pedestrians mainly rely on satellite positioning, and usually have poor positioning precision in urban canyons. As for vehicles, even though satellite positioning alone cannot provide sufficient precision, its combination with dead reckoning, map matching, and lane detection etc. help to keep positioning errors within a few meters [7,8]. In the future, it is expected that the positioning precision of vehicles will be further improved for autonomous driving, which will be much higher than that of pedestrians.

With the reasonable assumption that vehicle positions have much higher precision than that of pedestrians, our previous work proposed a method that uses vehicles as anchors in addition to GPS satellites, to improve pedestrian positioning precision [9,10]. In this method, a pedestrian device overhears signals (containing vehicle position) exchanged between vehicles (for the purpose of avoiding vehicle collisions) and calculates pedestrian–vehicle distance from the strength of the LoS (line of sight) wave using CSI (channel state information) computed from the signal. In addition, pedestrian–vehicle angle is estimated from the spatial phase difference of the LoS wave received by multiple antennas, and exploited in position computation [11]. However, this method requires a large number of antennas on pedestrian devices in order to improve the angle estimation precision, which is impractical due to the device size limitation.

To solve the aforementioned problem, in this paper, we propose a new method to precisely estimate radiation angle of the LoS wave with a single antenna, by tracking the temporal phase variation of the vehicle’s signal at a pedestrian device, caused by the Doppler shift in the presence of vehicle movement. On this basis, we further exploit angle information for pedestrian positioning. The proposed method is not affected by the number of antennas at a pedestrian device, which was a problem in the base method [11], because the proposed method acquires multiple CSI at multiple points (different receiving timings) of a single packet and estimates radiation angle using the phase variation of the LoS wave. Extensive evaluations via 3D (three-dimensional) ray tracing simulation confirm the effectiveness of the proposed method. Part of the idea on angle estimation and position computation was reported at ICVES 2019 (International Conference on Vehicular Electronics and Safety 2019) [12], and is extended in this paper from two aspects: 1) new angle estimation that considers pedestrian speed as well; and 2) smoothing the pedestrian position using a Kalman filter. In addition, more evaluation results are added to illustrate the impact of different factors (overlapping multipath waves under the limit of time resolution, interference signal, different frequency bands, and error in vehicle position).

In the rest of this paper, Section 2 reviews related research, including both the component techniques (ranging, angle estimation) and their combination for positioning. Then, Section 3 presents the proposed method, both the angle estimation and its application for pedestrian positioning. Section 4 describes the simulation setting for pedestrian positioning in urban canyons, and Section 5 analyzes the results and discusses factors affecting positioning precision. Finally, Section 6 concludes this paper and points out future works.

## 2. Related Research

### 2.1. Indoor Positioning via Radio Waves

In indoor environments, satellite signals can hardly be used for positioning due to the occlusion of buildings. As a result, short-range radio waves such as Wi-Fi are actively studied as alternative methods. In these methods, a target device receives a signal from each AP (access point) with known position of the AP, detects signal propagation distance and AoA (angle of arrival) from information, such as signal strength/phase, and on this basis computes its own position [13,14,15,16]. In addition, device-free methods [17,18,19,20] for detecting a person without any wearable device are also proposed, which monitor the changes in signal reception status between pairs of preset devices.

As shown on the left of Figure 1, a signal transmitted by a transmitter arrives at the receiver via different paths, and there are two types of waves: multipath waves that reflect/diffract on walls, floors, etc., and LoS wave that propagates via the LoS path without reflection. It is important to use only the LoS wave for the estimation of signal propagation distance and AoA because multipath waves will introduce extra propagation distance and phase compared with the LoS wave. RSSI (received signal strength indicator), usually used for distance estimation, is actually the overall strength involving both the LoS wave and all multipath waves, and its usage increases the error in distance estimation.

On the other hand, CSI, obtained from OFDM (orthogonal frequency division multiplexing; which transmits multiple frequency signals simultaneously and is a wireless modulation method used in Wi-Fi and LTE) modulated signals, can indicate signal strength and arrival timing of each path [21]. Therefore, the LoS and multipath waves can be separated according to the arrival timing. When the LoS wave exists, the wave received at the earliest time can be regarded as the LoS wave. For this reason, precision improvement can be expected by using CSI instead of RSSI for indoor positioning.

### 2.2. Estimation of Signal Propagation Distance

Signal strength of the LoS wave, L, attenuates according to the propagation distance, d, as follows:(1)$$L=a\log_{10}d+b,$$
where a and b are constants depending on antenna gain, etc., and can be derived by linear regression from sets of signal strength (L) and propagation distance (d) of the LoS wave. Then, signal propagation distance (d) can be estimated from the received signal strength (L) [13].

Another method for estimating propagation distance is to compute signal propagation time [14], as the difference of the signal reception timing and transmission timing, and its product with the light speed gives the propagation distance. However, it requires accurate time synchronization between a transmitter and a receiver.

### 2.3. Estimation of Angle of Arrival

AoA is often estimated by using an antenna array [15,16]. In order to improve communication quality, multiple antennas usually are exploited in the form of MIMO (multiple-input and multiple-output), and the antenna array can be used for AoA estimation as well. As shown in Figure 2, there is a slight difference in signal propagation distances from the transmitter to multiple antennas at the receiver, which causes phase difference. This phase difference depends on the angle of arrival θ and the distance l between adjacent antennas at the receiver. Therefore, the AoA is calculated from the phase (CSI) at each antenna using the MUSIC (multiple signal classification) method [22].

AoA estimation can also be performed by a directional antenna. By rotating a directional antenna, the direction of the strongest signal intensity is detected, and this direction is regarded as the AoA [23]. Using a beacon that rotates at high speed, a method which estimates AoA from Doppler shift caused by rotation has also been proposed [24]. However, these methods [23,24] require dedicated devices for angle estimation and cannot be applied to general mobile devices.

### 2.4. Base Positioning Method

In order to improve the positioning precision of pedestrians, a method using vehicles as the anchors for positioning was proposed, which combines the component techniques, LoS wave (Section 2.1), distance computation (Section 2.2), and AoA estimation (Section 2.3). In this method, a pedestrian overhears V2V signals (carrying vehicle position) exchanged between vehicles. The LoS wave from a vehicle sometimes may be blocked and does not reach the pedestrian (in a blind spot) due to obstacles. In this case, if the first wave obtained from the CSI is used as a LoS wave, a large error occurs in distance/AoA estimation. Therefore, the presence of a LoS wave is detected, and the first wave is used only when it is determined that a LoS wave does exist (Figure 3). Then, the strength of the LoS wave is used to compute the pedestrian–vehicle distance, and the pedestrian–vehicle angle is estimated by an antenna array. By combining signals from both GPS satellites and vehicles, this method improves pedestrian positioning precision.

This method requires receiving signals with a large number of antennas so as to improve the spatial resolution in AoA estimation. It is difficult, however, to increase the number of antennas in a mobile device due to its size limitation.

## 3. Proposed Method

### 3.1. Overview

In order to solve the problem of the base method [11] that requires that a pedestrian device have a large number of antennas for high-precision angle estimation, we propose a new angle estimation method and apply it for pedestrian positioning. This method explores the Doppler effect that the CSI changes at a receiver (pedestrian) even within the same signal because the transmitter (vehicle) moves while transmitting a signal. Therefore, angle estimation is performed by tracking the phase change of the LoS wave obtained from CSI of multiple locations (timings) in a single packet. Figure 4 shows an overview of the proposed method.

### 3.2. Prerequisites

It is assumed that each vehicle transmits its position and speed information periodically (e.g., every 100 ms) to nearby vehicles, in order to avoid collision accidents, by using onboard wireless devices for V2V communication. Such V2V communication has been put into practical use in Japan since 2015 [1]. In addition, this method assumes that pedestrians have mobile devices and can overhear signals from vehicles. The communication between vehicles and pedestrians can be realized by C-V2X (cellular vehicle to everything) standardized in 5th generation mobile communication.

### 3.3. Estimation of Pedestrian–Vehicle Angle

#### 3.3.1. Phase change due to pedestrian–vehicle movement

When a pedestrian overhears the signal transmitted by a vehicle at time *t*, the LoS wave’s signal strength $h_{t}$ is expressed by Equation (2).
(2)$$h_{t}=\alpha e^{-j2\pi \frac{d}{\lambda }+\phi_{0}},$$
where α is the amplitude of the signal, λ is the wavelength, d is the signal propagation distance, $\phi_{0}$ is the initial phase, and $e^{-j2\pi \frac{d}{\lambda }+\phi_{0}}$ contains the overall phase of the signal. After one minute Δt (e.g., 100 μs), the LoS wave’s signal strength $h_{t+\Delta t}$ is expressed by Equation (3). Since Δt is sufficiently small, it is assumed that the signal amplitude α does not change.
(3)$$h_{t+\Delta t}=\alpha e^{-j2\pi \frac{d+\Delta d}{\lambda }+\phi_{0}},$$
where Δd is the variation of signal propagation distance that occurs during Δt when the vehicle moves while transmitting the signal. From Equations (2) and (3), Δφ, the variation of signal phase of the LoS wave, due to the distance variation Δd, is computed as follows, which corresponds to the Doppler shift.
(4)$$\Delta \varphi =2\pi \cdot \frac{-\Delta d}{\lambda }.$$

#### 3.3.2. CSI Acquisition

Vehicles move at variable speeds and the time error between vehicles and the pedestrian device also changes, which may affect the phase computation. When acquiring CSI from multiple packets, accurate synchronization of transmission time is required, otherwise, each packet may have a different initial phase. This is difficult. Therefore, the proposed method obtains CSI from multiple locations within a single packet with time interval Δt, and uses the phase change of the LoS wave derived from the CSI. In OFDM signals used for V2V communication, CSI is calculated from the preamble near the beginning of a packet and updated per OFDM symbol thereafter by using the pilot signal in subsequent OFDM symbols. Then, the strength/phase of the LoS wave is acquired from each CSI (OFDM symbol) and the phase change Δφ is calculated. Here, Δt is converted to the number of OFDM symbols (OFDM symbol period is assumed to be the same at vehicles and the pedestrian device) and strength/phase of the LoS wave is acquired from two CSI (two OFDM symbols spaced by Δt). Then, their phase difference Δφ is calculated at the pedestrian device, which does not require time synchronization with vehicles.

#### 3.3.3. Calculation of the Signal Radiation Angle

As shown in the upper part of Figure 5, a vehicle moves at speed $v_{c}$ along the road with a direction angle $\theta_{c}$, and a pedestrian moves at speed $v_{p}$ with a direction angle $\theta_{p}$. As the pedestrian–vehicle distance is sufficiently long and Δt is very short, changes in angles and speeds during the short interval Δt are ignored. Considering the movement of both the vehicle and the pedestrian, their relative speed $v_{r}$ is calculated as follows:(5)$$v_{r}=\sqrt{{\left(v_{c}\cos\theta_{c}-v_{p}\cos\theta_{p}\right)}^{2}+{\left(v_{c}\sin\theta_{c}-v_{p}\sin\theta_{p}\right)}^{2}},$$
and the direction of $v_{r}$ is denoted as $\theta_{r}$. When the vehicle signal is received by the pedestrian, the variation of signal propagation distance Δd during the interval Δt is expressed by
(6)$$\Delta d=v_{r}\cdot \Delta t\cdot \cos\theta_{rp},$$
where $\theta_{rp}$ represents the direction of the pedestrian position with respect to $v_{r}$. Then, Δφ is expressed by
(7)$$\Delta \varphi =2\pi \cdot \frac{-v_{r}\cdot \Delta t\cdot \cos\theta_{rp}}{c}\cdot f_{c},$$
where $f_{c}$ is the center frequency of the signal and c is the speed of light. On this basis, $\theta_{rp}$ can be computed as follows:(8)$$\cos\theta_{rp}=\frac{-\Delta \varphi \cdot c}{2\pi \cdot v_{r}\cdot f_{c}\cdot \Delta t}.$$
After $\theta_{rp}$ is computed, the difference between $\theta_{r}$ and $\theta_{rp}$ leads to θ, the pedestrian angle with respect to the vehicle. This angle is called the radiation angle (because it defines the radiation direction of the LoS wave from the vehicle to the pedestrian) and is used for the calculation of pedestrian position.

#### 3.3.4. Comparison with Previous Methods

Figure 6 shows the comparison between the base method [11] and the proposed method. In the base method, the angle estimation precision depends on the number of antennas because the spatial phase difference of signals received by multiple antennas is used for angle estimation. In comparison, the proposed method uses the temporal phase change of the received signal generated by the Doppler shift due to vehicle movement. Therefore, its angle estimation is possible even with a single antenna, and its precision is not affected by the number of antennas. Another difference between the two methods is the type of estimation angle. The base method estimates the arrival angle of the LoS wave at a pedestrian, but the proposed method estimates the radiation angle of the LoS wave from a vehicle.

The proposed method is similar to a method [24] in which Doppler shift is used for angle estimation, but there are some essential differences. In contrast to the method [24] that uses the rotating beacon as a positioning reference point and directly measures/uses Doppler shift, the proposed method uses moving vehicles as reference points, and the Doppler shift is not measured directly, rather the signal phase change due to Doppler shift is used instead.

### 3.4. Estimation of Pedestrian–Vehicle Distance

As with the base method [11], the pedestrian–vehicle distance is calculated from the signal strength of the LoS wave using the method described in Section 2.2 that uses linear regression.

### 3.5. Presence Detection of the LoS Wave

Presence of the LoS wave is detected by the same method used in the base method [11], which is described in Figure 3. When a LoS wave does not exist, the signal will not be used for the position calculation.

### 3.6. Positioning Calculation

In the positioning calculation, a Kalman filter is used to reduce error by smoothing the position.

#### 3.6.1. Initial Setting

The pedestrian position x, y,z, moving speed $v_{x},v_{y},v_{z}$, and distance error due to clock drift δ, are set as state X, and k satellite–pedestrian distances $d_{s1},\ldots ,d_{sk}$, n pedestrian–vehicle distances $d_{c1},\ldots ,d_{cn}$, n pedestrian-–vehicle angle $\theta_{1},\ldots ,\theta_{n}$, and pedestrian moving speed $s_{x},s_{y},s_{z}$ are set as observation value Y.
(9)$$X={\left[x,\;y,z,v_{x},v_{y},v_{z},\delta \right]}^{T},$$
(10)$$Y={\left[d_{s1},\ldots ,d_{sk},d_{c1},\ldots ,d_{cn},\theta_{1},\ldots ,\theta_{n},s_{x},s_{y},s_{z}\right]}^{T}.$$ State ***X*** is the unobservable state and computed from the observation value ***Y*** measured by the pedestrian device.

#### 3.6.2. Prediction Step

At a timestep t, the current state $X_{t}^{-}$ is predicted from the previous state $X_{t-1}^{+}$ at timestep t−1, by using the state transition model F, as follows:(11)$$X_{t}^{-}=FX_{t-1}^{+},$$
(12)$$F=\left[\begin{array}{ccccccc} I & & & \Delta T\cdot I & & & \\ & I & & & \Delta T\cdot I & & \\ & & I & & & \Delta T\cdot I & \\ & & & I & & & \\ & & & & I & & \\ & & & & & I & \\ & & & & & & I \end{array}\right].$$
ΔT is the difference between adjacent timesteps. As for the state covariance P, its value $P_{t}^{-}$ at the current timestep t is predicted using the estimated value $P_{t-1}^{+}$ at the previous timestep t−1 and the process noise covariance Q.
(13)$$P_{t}^{-}=FP_{t-1}^{+}F^{T}+Q.$$

The Kalman gain K is obtained using these predicted values, the observation model H and the measurement noise covariance R of observation value Y.
(14)$$K_{t}=P_{t}^{-}H^{T}{\left(HP_{t}^{-}H^{T}+R\right)}^{-1}.$$
H is composed of unit direction vectors from the pedestrian to each satellite/vehicle calculated using $X_{t}^{-}$ and satellite/vehicle position.

#### 3.6.3. Update Step

The final state $X_{t}^{+}$ is computed as the sum of the predicted state $X_{t}^{-}$, and the product of the Kalman gain $K_{t}$ and the difference between the observation value $Y_{t}$ and the predicted distance $\rho \left(X_{t}^{-}\right)$, and gives the positioning result at this timestep.
(15)$$X_{t}^{+}=X_{t}^{-}+K_{t}\left(Y_{t}-\rho \left(X_{t}^{-}\right)\right).$$
The state covariance is also updated, by using Kalman gain and the predicted value $P_{t}^{-}$.
(16)$$P_{t}^{+}=\left(I-K_{t}H_{t}\right)P_{t}^{-}.$$
At the next timestep, the time is increased to t+1 and the same process is repeated.

## 4. Simulation Evaluation

The performance of the proposed method is verified by simulation. In the simulation, radio wave propagation simulation software “RapLab” (provided by Kozo Keikaku Engineering Inc.) [25] is used to perform 3D ray tracing simulation of signals from vehicles and GPS satellites to obtain signal path information. Then, on this basis, the estimation of pedestrian-–vehicle distance/angle and the position calculation are performed by the numerical analysis software “MATLAB” (provided by Math Works Inc.) to evaluate the performance of angle estimation and positioning precision. The evaluation of the base method [11] is also performed for comparison.

### 4.1. Simulation Conditions

Assuming an urban environment, 3D building data near the Ginza 4-chome intersection (provided by NTT Data Corporation), shown in Figure 7a, is used for the simulation. GPS satellites are placed based on the actual ephemeris data. Locations of a moving pedestrian and moving vehicles are shown on the map, and the aerial photograph of the same area acquired by Google Earth is shown in Figure 7b.

Table 1 shows the detailed (default) simulation conditions. In the simulation, 200 scenes with a time interval 0.1 s are created to emulate the movement of vehicles and pedestrians. The setting of the transmission signal of vehicles is determined according to the standard of V2V communication [1]. After reading building data with RapLab, the pedestrian, vehicles, and GPS satellites are placed on the map, and radio wave propagation information is obtained by 3D ray tracing. In the ray tracing, signal obstruction/reflection by the terrain (ground, building, etc.) is simulated.

After the ray tracing simulation, for each scene, the presence detection of the LoS wave, the estimation of pedestrian–vehicle distance/angle, and the position calculation are performed by MATLAB using the acquired signal propagation information. The pedestrian–vehicle distance is predicted by a linear regression model defined in Equation (1) when given the LoS wave signal strength as input. Satellites with elevation angles no more than 15 degrees and vehicles with distance longer than 40 m or without a LoS wave are not used in the calculations, because they are considered to have large ranging errors.

### 4.2. Emulation of Time Resolution

In a real environment, a receiver has a limited time resolution. Therefore, multiple waves received within an interval shorter than the time resolution of the receiver cannot be separated. As a result, the signal strength/phase becomes a combined value of multiple waves that overlap together. When estimating the distance or angle using the LoS wave, errors (multipath errors) occur if the LoS wave is overlapped by subsequent multipath waves.

RapLab provides ideal CSI, where all paths are separated. In order to perform a realistic simulation considering the limitation of time resolution, the CSI obtained by RapLab is converted to an equivalent at a time resolution of 50 ns, corresponding to that of a general wireless local area network (running with 20 MHz bandwidth). This is done by combining signal strength values of multiple waves that overlap together, using Equation (17), where $\alpha_{k}+i\beta_{k}$ represents the complex strength of the kth wave and n is the number of overlapping waves within the interval corresponding to the time resolution (Figure 8). After this process, the combined CSI is used for position calculation.
(17)$$\alpha =\left(\alpha_{1}+\alpha_{2}+\cdots +\alpha_{n}\right)+i\left(\beta_{1}+\beta_{2}+\cdots +\beta_{n}\right).$$

### 4.3. Simulation of Thermal Noise on Signal Propagation Path

CSI in the previous section is obtained without noise. To better emulate the real wireless channel, we simulated the whole signal propagation path considering thermal noise.

In this process, the transmission (Wi-Fi OFDM) signal passes the wireless channel (using the CSI created in Section 4.2). When it is received at the receiver side, noise corresponding to specified SNR (signal-to-noise ratio) is added to the signal. Then, CSI is estimated from the OFDM signal (with noise). Here, SNR is calculated from the received signal power and the thermal noise power, where the former is calculated by 3D ray tracing and the latter is calculated by Equation (18).
(18)$$P_{n}=\frac{hf_{c}B}{e^{\frac{hf_{c}}{kT}}-1}\approx kTB\;\left(hf_{c}\ll kT\right).$$
where $f_{c}$ and B are the center frequency and bandwidth of the signal, respectively, T is temperature, h is Plank’s constant, and k is Boltzmann’s constant. Usually αkTB is used as the noise power with a constant α, so that the noise level matches the real environment. In the simulation, α is 10, temperature is 25 °C, and bandwidth is 20 MHz.

### 4.4. CSI Acquisition Settings

In the proposed method, a single antenna is used for signal reception, and CSI is obtained from two locations with a time interval Δt. The phase variation in CSI increases with the length of Δt. If the phase variation is small, the angle precision will decrease due to the influence of noise. Considering this fact, it is expected that the angle estimation precision will improve as the acquisition interval of CSI increases. In the simulation, this interval is set to be long enough, Δt=100 μs, but not exceed the packet transmission time (150 μs to 300 μs) decided by the V2V communication standard [1]. In order to acquire CSI after the time interval Δt, additional 3D ray tracing simulation that reflects vehicle/pedestrian movement during Δt is performed at each scene.

In the base method [11], 4, 6, or 8 antennas are used for signal reception, and the CSI is obtained from the received signal for each antenna.

## 5. Simulation Results and Discussion

### 5.1. Results for Angle Estimation

#### 5.1.1. Comparison with the base method

Figure 9 shows the CDF (cumulative distribution function) of angle error and Table 2 shows the average result. The CDF figure confirms that the angle error in the proposed method is smaller than that in the base method as a whole. In particular, the percentage of relatively large angle errors of 1 degree or more is small in the proposed method. On average, the proposed method reduces the angle error by about 70%, compared to the base method with eight antennas.

From the analysis of 3D ray tracing results, the reason why the proposed method has higher angle estimation precision than the base method is because it is less affected by multipath errors. Due to the influence of the time resolution of the receiver, the LoS wave obtained from CSI is overlapped by the multipath waves immediately following it. This wave is different from the true LoS wave, so hereafter, it is called pseudo LoS wave. For the aforementioned reason, multipath errors occur when angle estimation is performed using pseudo LoS wave. In the base method, since the AoA of the received signal is estimated, the multipath error mainly occurs due to the arrival angle difference between the LoS wave and the overlapping multipath waves. In comparison, the proposed method estimates the radiation angle of the signal, and the multipath error mainly occurs due to the radiation angle difference between the LoS wave and the overlapping multipath waves.

Figure 10a shows the LoS wave and the overlapping multipath waves on the map, and Figure 10b shows the aerial photograph of the same location. In many cases, pedestrians move on the sidewalk near the building, and vehicles move on the road with a longer distance from the building. Therefore, the radiation angle difference between the LoS wave and the overlapping multipath waves from the vehicle is small, but the arrival angle difference is large. In the base method, the angle error increases due to the large difference in arrival angle, and the angle error is 29.18 degrees in the scenario shown in Figure 10a. In comparison, in the proposed method, the angle estimation is not significantly affected because the radiation angle difference is small, and the angle error is only 2.99 degrees.

#### 5.1.2. Influence of Pedestrian–Vehicle Distance

Figure 11 shows the average value of the radiation angle estimation error for each pedestrian–vehicle distance in the proposed method. It is clear that the angle error is minimum when the distance is in the range (0, 8) m, and tends to increase with the distance. This is because the increase in the signal propagation distance decreases the signal strength, which degrades the angle precision in the presence of noise.

In fact, the estimation error of distance also increases with distance. Therefore, in this simulation, only vehicles with a pedestrian–vehicle distance no more than 40 m are used for positioning. When there are sufficient vehicles around the pedestrian, it is possible to further improve the positioning precision by using nearer vehicles.

#### 5.1.3. Influence of Interference Wave

Many vehicles run on the road and contend to broadcast their position information. Then, transmission collisions occur unavoidably due to either the hidden terminal problem or simultaneous transmissions. As a result, signal interference sometimes occurs. When a pedestrian receives a V2V communication signal and an interference wave is present, the precision of the angle estimation is degraded.

In the following, the impact of interference on the proposed method is investigated. In the evaluation, another OFDM signal created from random data is added to the received signal as an interference, together with the noise. The amplitude of the interference is adjusted according to the preset SIR (signal-to-interference ratio).

Figure 12 shows the average angle error of the proposed method for each SIR. Generally, the angle error due to the interference increases greatly when the SIR gets smaller. In the case where SIR is 10 dB, the average angle error is about 17 degrees. On the other hand, when the SIR is greater than 30 dB, the influence is small, and the precision of the angle estimation is almost the same as the case without interference. Therefore, it can be assumed that the influence of the interference can be neglected when SIR is 30 dB or higher.

#### 5.1.4. Influence of Signal Transmission Frequency

In the above evaluation, the simulation is performed in the 700 MHz band, but the 5.9 GHz band is also adopted in the V2V communication standard. Here, we investigate whether the angle estimation of the proposed method is effective in other frequency bands. Table 3 shows the average value of the angle errors in the 2.4 GHz and 5 GHz bands that are used in general Wi-Fi, and Figure 13 shows the CDF of angle error. In this result, the angle error is smallest in the 700 MHz band, and increases in the 5 GHz and 2.4 GHz bands.

There are mainly two factors that affect the angle estimation in the proposed method. As shown in Equation (7), the phase difference that occurs in a short time is proportional to the center frequency of the signal. Therefore, as the frequency increases, the phase variation increases and relative measurement error decreases. On the other hand, the frequency also affects the distance attenuation of the signal, and the attenuation increases in proportion to the square of the frequency. As a result, even at the same propagation distance, the received signal strength (SNR) decreases as the frequency increases. Here, the angle estimation precision does not change largely when the frequency is increased, probably because the gain in phase difference is cancelled by the degradation in the SNR.

### 5.2. Results for Positioning

#### 5.2.1. Comparison with the Base Method

Figure 14 shows the CDF of the horizontal positioning errors, and Table 4 shows the average value, for the proposed method and the base method. In the CDF figure, some large errors occur in the proposed method and the base method. These large errors actually occur immediately after the start of positioning, when the results are not well smoothed by the Kalman filter yet. As for the average positioning error, using only GPS results in the largest error (12.08 m). The average positioning error of the proposed method is less than 1 m and is reduced by about 69% compared with that of the base method using eight antennas.

#### 5.2.2. Influence of the Number of Vehicles

It is known that positioning performance depends on the number of used anchors. For evaluating the precision of the proposed method when the number of surrounding vehicles is small, the number of vehicles used in each positioning scene is changed, and the positioning error is investigated. In this simulation, vehicles used for positioning are randomly selected from the available vehicles in each scene. In order to confirm the superiority of using angle in addition to distance, we compare the results of positioning using pedestrian–vehicle distance and GPS, but without using angle.

Figure 15 shows the average horizontal positioning error. In both methods, the positioning error is large when the number of vehicles is small, but the error is reduced almost by half when using the angle information in the proposed method. From this result, it can be seen that if the target positioning error is within 1 m, the number of vehicles should be about five or more. In the simulation, the number of vehicles in each scene is equal to 15 on average, but in the proposed method even when the number of vehicles is eight, the result is still almost the same as the one in Section 5.2.1 where positioning is performed with the maximum number of available vehicles. Therefore, it is thought that the proposed method can work well even with a small number of vehicles.

#### 5.2.3. Influence of Position Error of Vehicles

In the previous simulations, the position of each vehicle is assumed to be correct without error. However, positioning errors, although small, actually also occur in vehicles. Here, random errors are generated at the vehicle position in a normal distribution based on the preset average error value, and the influence on the positioning precision in the proposed method is evaluated.

Figure 16 shows the pedestrian’s average positioning error under different vehicle position errors. It is clear that as the position error of vehicles increases, the positioning error of pedestrians by the proposed method does not increase as much. This is because the error is smoothed by the Kalman filter. In this simulation, a random error is given to each vehicle, using multiple vehicles and the continuous time measurements together with a Kalman filter smoothens the effect of the vehicle position errors, so the positioning error of the pedestrian is not increased significantly. On the other hand, in the case where vehicle position errors are biased, it is difficult to reduce the effect of the vehicle position error by smoothing, and its verification is left as future work.

#### 5.2.4. Influence of Interference Wave

As in Section 5.1.3, the same SIR is set for all vehicle signals, and the impact of the interference signal on the positioning precision is investigated. Figure 17 shows the average horizontal positioning error under different SIR. Similar to the results of angle estimation, when the SIR is 30 dB or higher, the interference has almost no impact on the positioning precision. Further, when the SIR decreases to 10 dB, the positioning error increases to about 2 m.

In the simulation, the interference signal is assumed to be in all V2V transmissions, but in V2V communication, transmission collision avoidance is performed by carrier sense (CSMA/CA, carrier sense multiple access with collision avoidance). This removes most interference although not all. In the presence of hidden terminals, if SIR is low, the packet will not be correctly received, and the sender vehicle will not be used as an anchor. For this reason, the probability of the interference signal is much lower in the actual environment. Therefore, the influence of interference signal on the positioning precision is expected to be much smaller.

#### 5.2.5. Influence of Signal Transmission Frequency

Table 5 shows the average of the horizontal positioning error when the transmission signal frequency of the vehicle is changed (Section 5.1.4) and Figure 18 shows the CDF. Similar to the result of angle estimation, the error is smallest when the frequency is 700 MHz. On the other hand, the error is smaller in the 2.4 GHz band than in 5 GHz band, and the difference between results in 2.4 GHz/5 GHz and those in 700 MHz is relatively larger compared with the result of angle estimation. This is because the transmission frequency affects not only the angle estimation but also the distance estimation. Signal attenuation gets larger with the frequency, so the precision of distance estimation will decrease at higher frequency. As a result, the higher the frequency, the larger the positioning error, because the precision of both the angle estimation and the distance estimation is degraded.

### 5.3. Summary

Investigations on the impacts of multipath waves, pedestrian–vehicle distances, interference signals, and frequency bands show that the accuracy of the angle estimation in the proposed method is decreased due to the degradation of the received signal (low SINR). Extensive evaluations show that the main factor of the angle error is the multipath wave (under the constraint of time resolution), and the environment that causes large angle errors is different between the proposed method and the base method due to the difference in the type of the estimated angle. The proposed method is less susceptible to the impacts of multipath waves reflected from buildings near pedestrians, however, this leads to large errors in the base method. It is a typical case that pedestrians walk near roadside buildings, so the proposed method usually outperforms the base method. Although multiple antennas, when available, do provide extra measurement. In the future, we will also try to use multiple antennas in the proposed method.

Compared with the base method, the proposed method generally achieves higher positioning precision. Although the number of available vehicles affects the precision of pedestrian position in the proposed method, the use of angle information significantly improves the positioning precision when the number of vehicles is small.

## 6. Conclusions

In this paper, in order to better prevent pedestrian accidents by P2V communication, we proposed a high-precision angle estimation method and applied it to achieve high-precision pedestrian positioning, in which a pedestrian mobile device overhears V2V signals containing vehicle location information, and estimates pedestrian–vehicle distance/angle information for position computation. CSI is obtained from multiple locations of a single packet received at a single antenna with different receiving timings, and the angle is estimated from the phase variation of the LoS wave obtained from the CSI. This solves the problem that the pedestrian device needs a large number of antennas for highly accurate angle estimation in the base method.

Three-dimensional ray tracing simulation confirmed that the angle estimation error could be reduced from 13 degrees to 3.9 degrees and the positioning error could be reduced from 2.49 m to 0.78 m compared to the base method using eight antennas in the receiver.

Future works include improving the positioning precision by acquiring CSI from three or more points in a single packet, joint use of temporal and spatial phase differences for angle estimation, and verifying the precision of the proposed method in more scenarios.

## Figures and Tables

**Figure 1 sensors-20-01430-f001:**
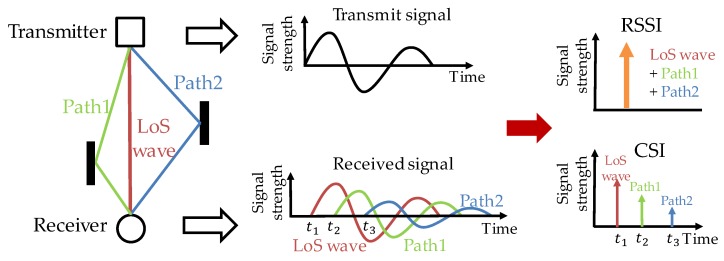
Signal propagation path and RSSI/CSI (received signal strength indicator/channel state information).

**Figure 2 sensors-20-01430-f002:**
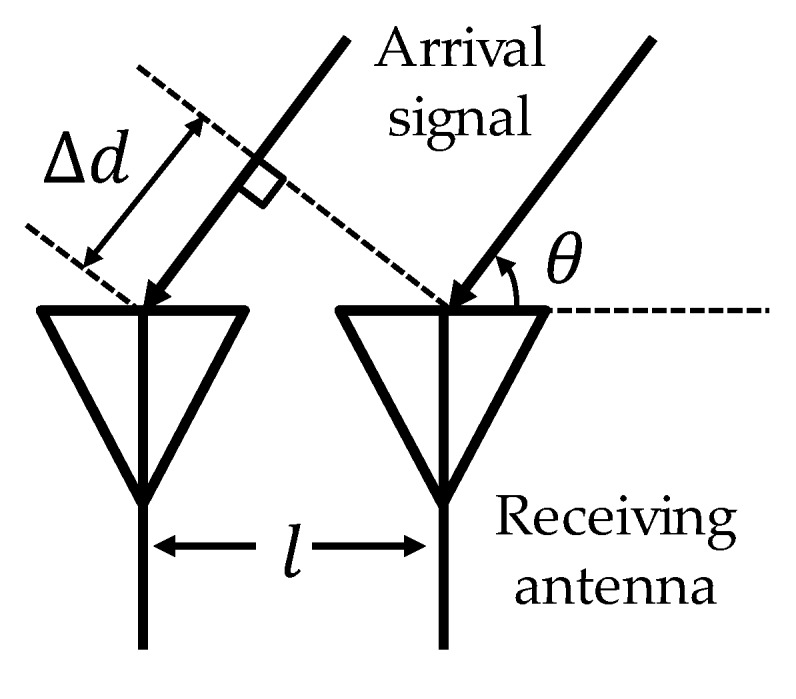
Signal propagation distance difference between two receiving antennas.

**Figure 3 sensors-20-01430-f003:**
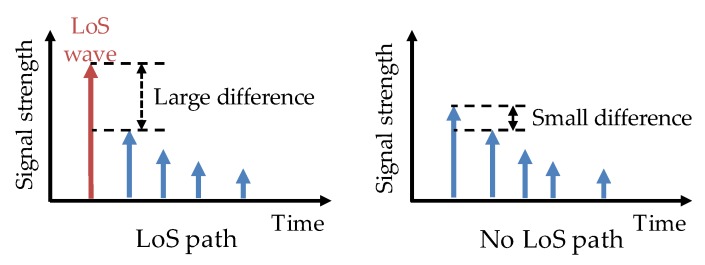
Presence detection of the LoS (line of sight) wave. The difference between the signal strengths of the first wave and the second one is large when there is a LoS path, and small otherwise.

**Figure 4 sensors-20-01430-f004:**
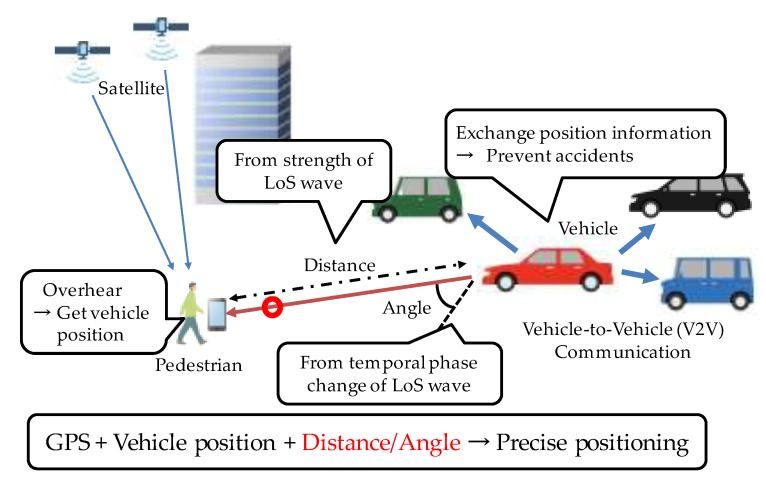
Overview of pedestrian positioning in the proposed method. GPS = Global Positioning System.

**Figure 5 sensors-20-01430-f005:**
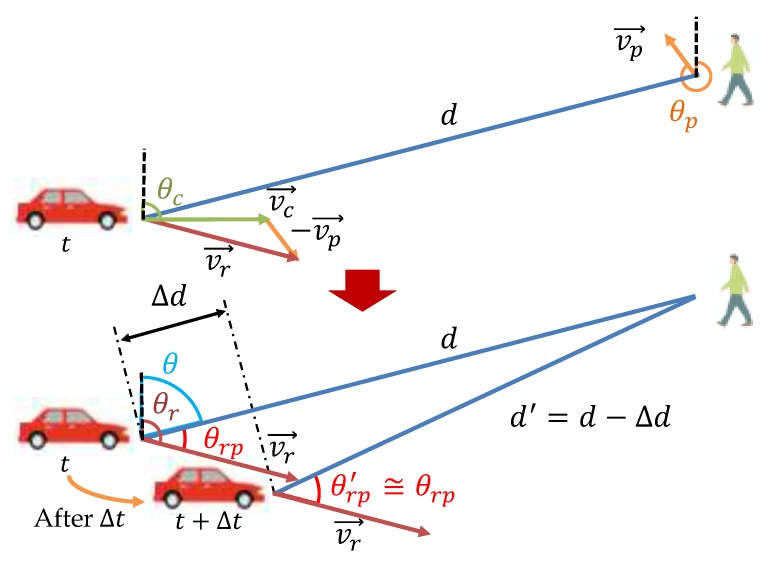
Change of signal propagation distance during Δt

**Figure 6 sensors-20-01430-f006:**
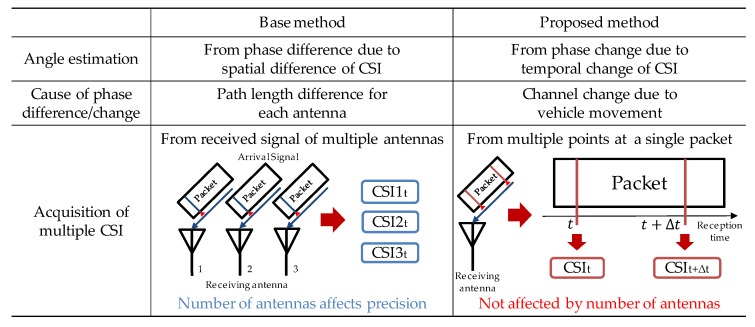
Comparison between the base method [11] and the proposed method.

**Figure 7 sensors-20-01430-f007:**
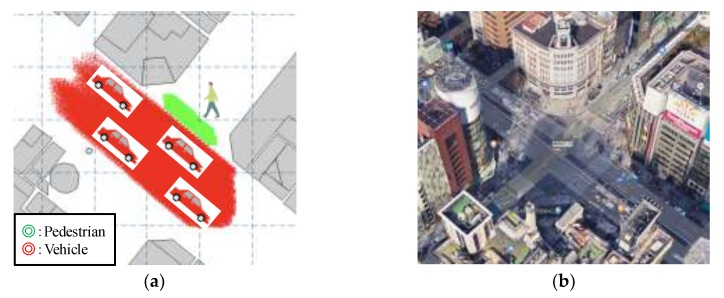
Evaluation scenario. (**a**) Placement of a moving pedestrian and vehicles. (**b**) The aerial photograph of the simulation map (Ginza 4-chome intersection).

**Figure 8 sensors-20-01430-f008:**
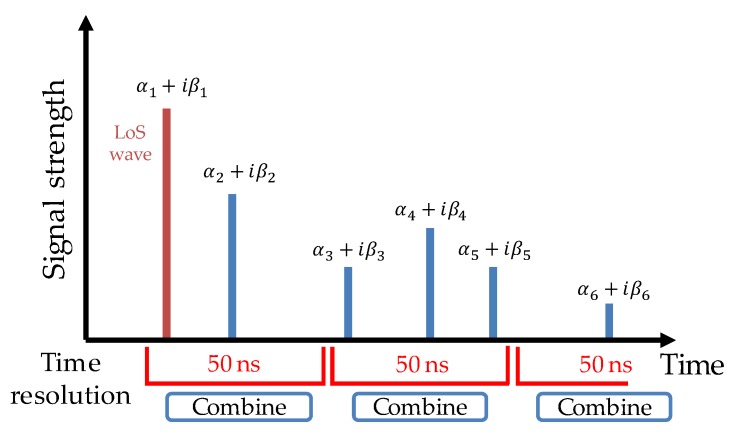
Simulation of time resolution.

**Figure 9 sensors-20-01430-f009:**
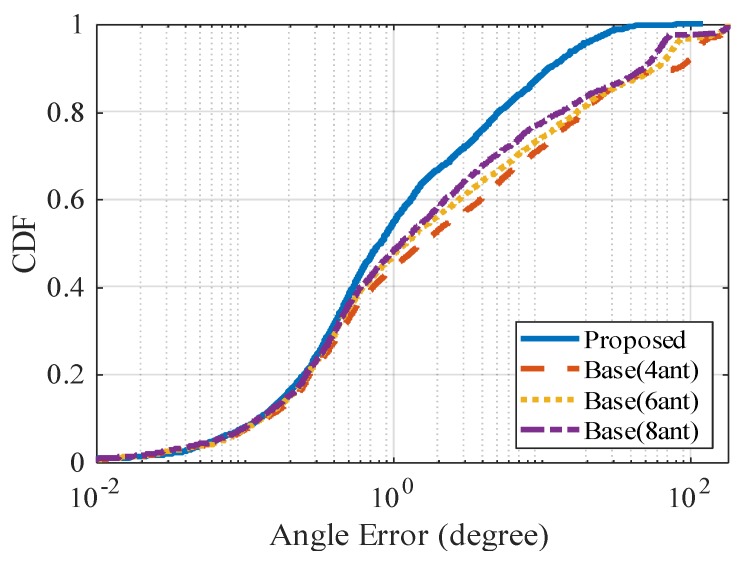
CDF (cumulative distribution function) of angle error.

**Figure 10 sensors-20-01430-f010:**
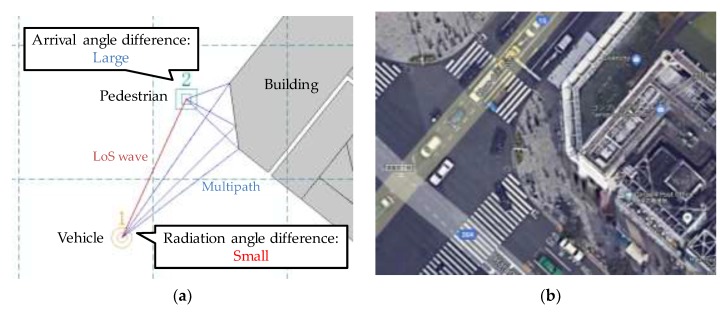
Signal propagation from a vehicle to a pedestrian. (**a**) Radiation and arrival angles of multipath waves combined with a LoS wave. (**b**) The aerial photograph of the same location (acquired by Google Earth).

**Figure 11 sensors-20-01430-f011:**
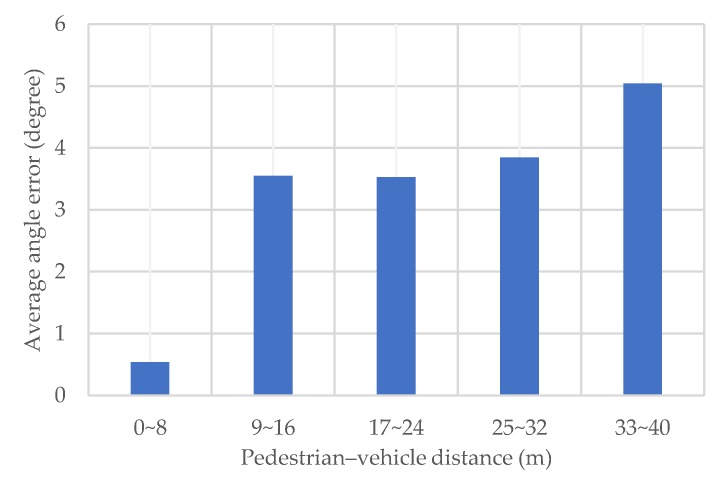
Average angle error for each pedestrian–vehicle distance.

**Figure 12 sensors-20-01430-f012:**
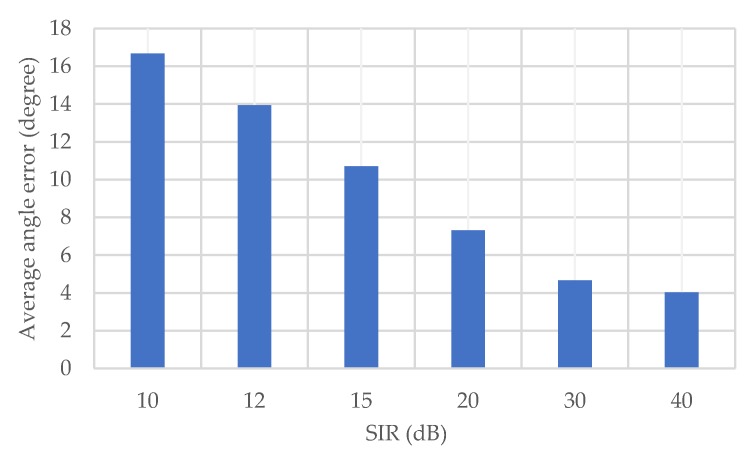
Average angle error for each SIR (signal-to-interference ratio).

**Figure 13 sensors-20-01430-f013:**
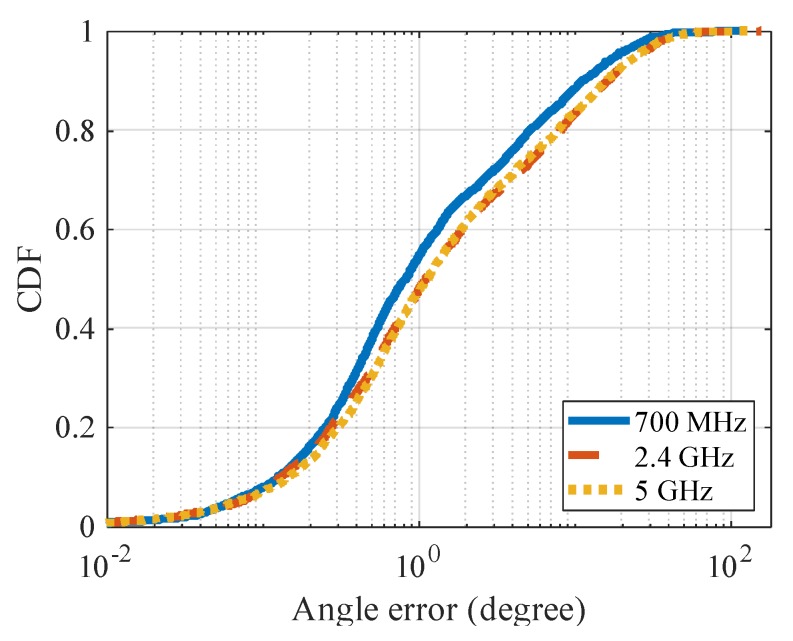
CDF of angle error for each frequency.

**Figure 14 sensors-20-01430-f014:**
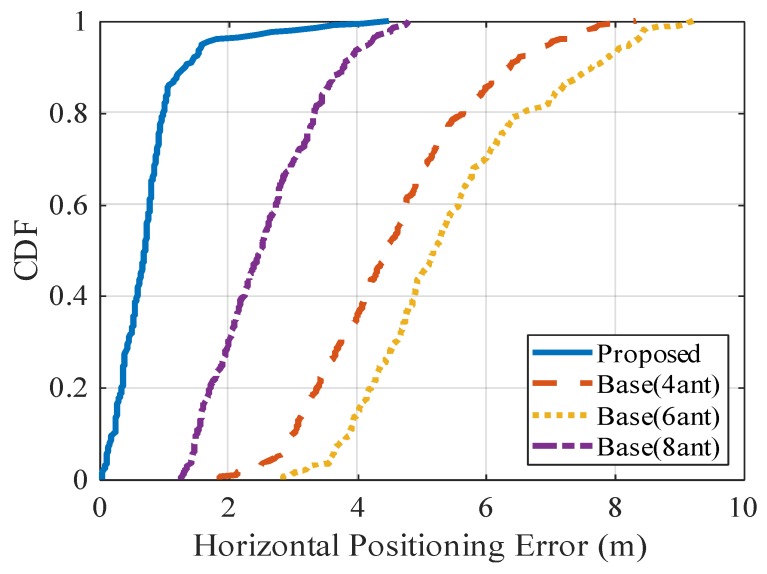
CDF of horizontal positioning error.

**Figure 15 sensors-20-01430-f015:**
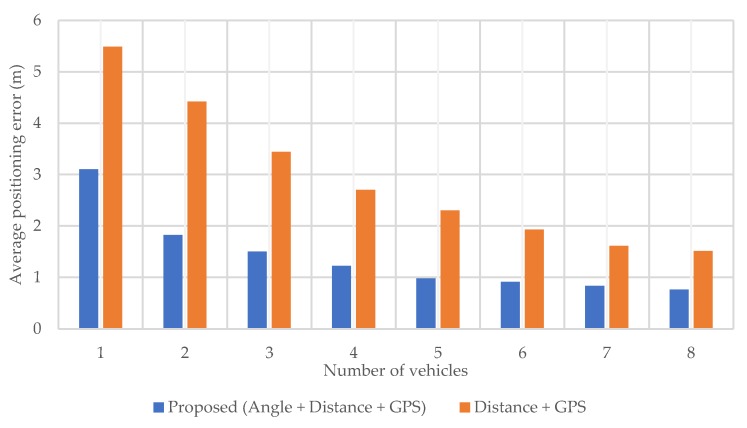
Average horizontal positioning error for each number of vehicles.

**Figure 16 sensors-20-01430-f016:**
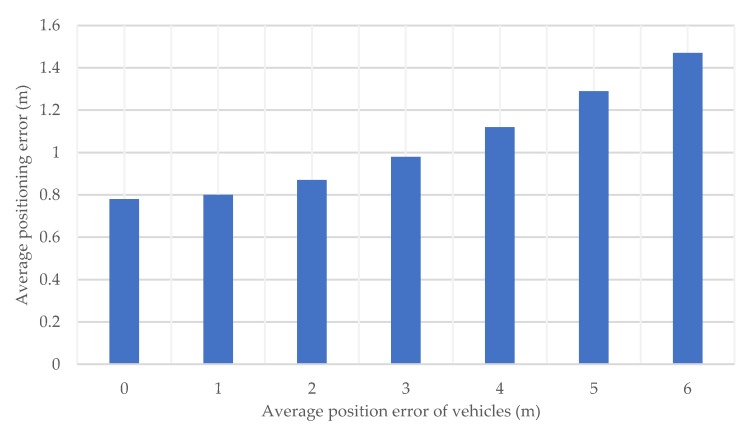
Average horizontal positioning error for each average position error of vehicles.

**Figure 17 sensors-20-01430-f017:**
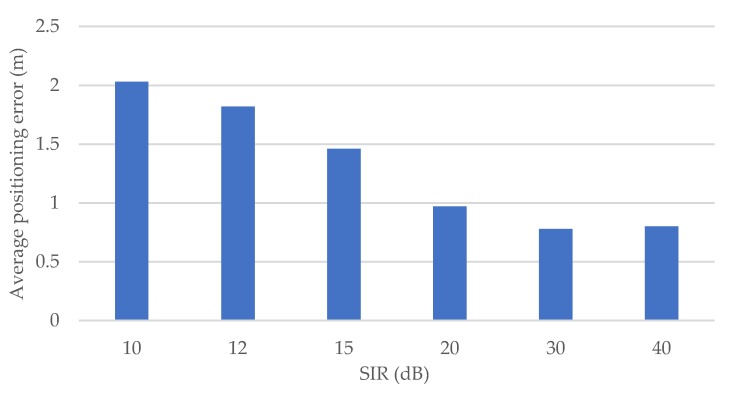
Average horizontal positioning error for each SIR.

**Figure 18 sensors-20-01430-f018:**
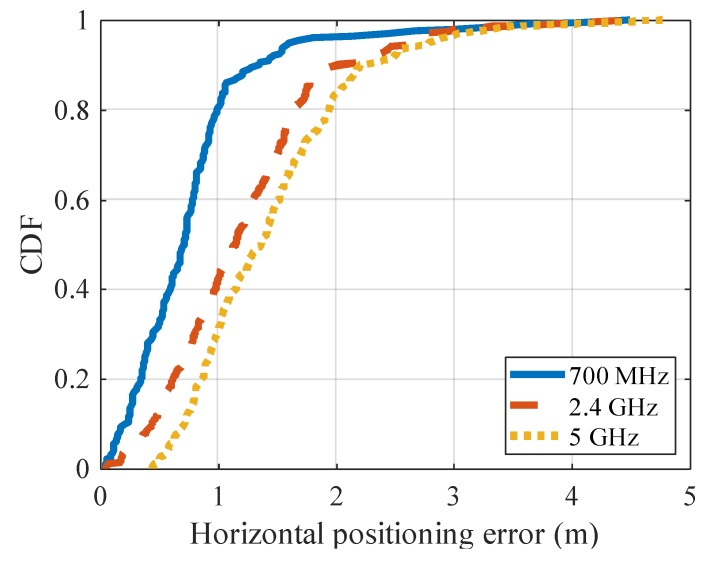
CDF of horizontal positioning error for each frequency.

**Table 1 sensors-20-01430-t001:** Simulation conditions.

Simulator	RapLab (3D ray Tracing) MATLAB (Distance/Angle/Position Calculation)
Trial conditions	200 times every 0.1 s from 2019/1/1 PM1:00
Vehicle position/number	2 lanes on each side, move at 60 km/h, randomly arranged with a head way distance of 5 to 30 m, 15 vehicles on average
Pedestrian position	1 person, moves on the pedestrian crossing (4 km/h)
Satellite	GPS satellites (elevation angle 15 degrees or more)
Vehicle signal	Center frequency: 700 MHz, Transmit power: 20 dBm, transmit interval: 0.1 s
3D raytracing	Maximum number of signal reflection: 1 Maximum number of signal diffraction: 1
Vehicle position error	None

**Table 2 sensors-20-01430-t002:** Average of angle error (degree).

	Proposed	Base		
		8 antennas	6 antennas	4 antennas
Angle error	3.90	13.00	15.56	18.70

**Table 3 sensors-20-01430-t003:** Average of angle error for each frequency band (degree).

	700 MHz	2.4 GHz	5 GHz
Angle error	3.90	5.36	5.23

**Table 4 sensors-20-01430-t004:** Average of horizontal positioning error (m).

	Proposed	Base		
		8 antennas	6 antennas	4 antennas
Positioning error	0.78	2.49	5.46	4.58

**Table 5 sensors-20-01430-t005:** Average of horizontal positioning error for each frequency band (m).

	700 MHz	2.4 GHz	5 GHz
Positioning error	0.78	1.23	1.46

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
