# Peer review of "Radiation Angle Estimation and High-Precision Pedestrian Positioning by Tracking Change of Channel State Informationâ€"

_sensors, 2020, doi:10.3390/s20051430_

Round 1
Reviewer 1 Report
The authors present an interesting paper about pedestrian position estimation by tracking CSI parameters. You present a novel approach that seem to me it could be promising. However, some improvements should be implemented, namely:
1.- authors refer that this article is an update to a conference paper (ICVES'19), but it is not clear in the text (introduction) what the changes / improvements were actually made. Is it just the application of the kalman filter to soften the pedestrian's position? Are there any new approaches in this work?
2.- in the description of the methodology used, the authors refer that (pag11) the new method is less affected by multipath errors? Is this statement a conclusion of the tests, or is it obtained based on the simplifications presented (pag5-6)?
3.- during the comparison with previous methods, the authors essentially propose a change from the "spatial domain" to the "time domain", does this mean that the studied signals are more stable temporally than spatially? or there are other reasons.
4.- the estimation remains based on "angles", right? it all comes down to changing the calculation end-point, ie Pedestrian (Arrival angle) or Vehicle (Radiation angle)? obtaining the latter should be better explained.
5.- furthermore, the test conditions should be better defined and differentiated. For example, only LoS is analyzed? what if there is an obstacle in the meantime?
Finally, congratulations on your work.
Author Response
Point 1: authors refer that this article is an update to a conference paper (ICVES'19), but it is not clear in the text (introduction) what the changes / improvements were actually made. Is it just the application of the kalman filter to soften the pedestrian's position? Are there any new approaches in this work?
Response 1: Thanks for this comment.
This manuscript extends its conference version from two aspects.
About the method itself, pedestrian speed, being small compared with vehicle speed, was neglected in the conference version, but is taken into account in this manuscript. In addition, a Kalman filter is used for smoothing the pedestrian position.
As for the evaluation, we extensively investigate the impact of different factors.
We have added more details to explain this more clearly in the introduction part, as follows.
Part of the idea on angle estimation and position computation has been reported in ICVES’19 [12], and is extended in this paper from two aspects; 1) new angle estimation that considers pedestrian speed as well, and 2) smoothing the pedestrian position using a Kalman filter. In addition, more evaluation results are added to illustrate the impact of different factors (overlapping multipath waves under the limit of time resolution, interference signal, different frequency bands, and error in vehicle position).
Point 2: in the description of the methodology used, the authors refer that (pag11) the new method is less affected by multipath errors? Is this statement a conclusion of the tests, or is it obtained based on the simplifications presented (pag5-6)?
Response 2: Thanks for this comment.
The statement that the new method is less affected by multipath errors is a conclusion drawn from the analysis of simulation results (Fig.9-10). We have clarified this in Section 5.1.1, as follows.
From the analysis of 3D ray tracing results, the reason why the proposed method has higher angle estimation precision than the base method is considered that it is less affected by multipath errors.
Point 3: during the comparison with previous methods, the authors essentially propose a change from the "spatial domain" to the "time domain", does this mean that the studied signals are more stable temporally than spatially? or there are other reasons.
Response 3: Thanks for this comment.
First, as mentioned in Section 2.4. and 3.1., initially the purpose of changing from spatial domain to time domain is to resolve the problem of the base method that requires a pedestrian device be equipped with an antenna array to get spatial phase difference of the signal. In comparison, temporal phase change of the signal could be acquired by even with a single antenna.
Simulation results also show the superiority of the proposed method over the base method with multiple antennas in angle estimation and positioning. This is because the proposed method using temporal phase difference (caused by Doppler shift) in angle estimation, which is less susceptible to multipath error.
But multiple antennas, when available, do provide extra measurement. In the future, we will investigate the joint use of spatial and temporal phase difference in angle estimation.
We have added a paragraph at the end of Section 5 to summarize the impact of different factors on the proposed method, and explain when the proposed method works well, and why it outperforms the base method, as follows.
5.3. Summary
Investigations on the impacts of multipath waves, pedestrian-vehicle distances, interference signals, and frequency bands show that the accuracy of the angle estimation in the proposed method will be decreased due to the degradation of the received signal (low SINR). Extensive evaluations show that the main factor of the angle error is the multipath wave (under the constraint of time resolution), and the environment that causes large angle errors is different between the proposed method and the base method due to the difference in the type of the estimated angle. The proposed method is less susceptible to the impacts of multipath waves reflected from buildings near pedestrians, which, however leads to large errors in the base method. It is a typical case that pedestrians walk near roadside buildings, so the proposed method usually outperforms the base method. But multiple antennas, when available, do provide extra measurement. In the future, we will also try to use multiple antennas in the proposed method.
Compared with the base method, the proposed method generally achieves higher positioning precision. Although the number of available vehicles affects the precision of pedestrian position in the proposed method, the use of angle information significantly improves the positioning precision when the number of vehicles is small.
We also modified the description about future work, as follows.
Future works include improving the positioning precision by acquiring CSI from three of more points in a single packet, joint use of temporal and spatial phase differences for angle estimation, and verifying the precision of the proposed method in more scenarios.
Point 4: the estimation remains based on "angles", right? it all comes down to changing the calculation end-point, ie Pedestrian (Arrival angle) or Vehicle (Radiation angle)? obtaining the latter should be better explained.
Response 4: Thanks for this comment.
The estimation of pedestrian position depends on “angle with respect to vehicles” besides “distance to vehicles”. As for the angle, the base method uses multiple antennas, and computes angle of arrival directly. In comparison, in the proposed method, the angle is defined with respect to the vehicle, and is called radiation angle. It is computed by using the temporal phase difference at an antenna, as is described in Sec.3.3.3.
We have modified the sentence below equation (8) to clarify the definition of radiation angle.
After θ_rp is computed, the difference between θ_r and θ_rp leads to θ, the pedestrian angle with respect to the vehicle. This angle is called radiation angle (because it defines the radiation direction of the LoS wave from the vehicle to the pedestrian), and is used for the calculation of pedestrian position.
We have added sentences about the difference between the proposed method (using radiation angle) and the base method in Section 3.3.4, as follows.
Another difference between the two methods is the type of the estimation angle. The base method estimates arrival angle of the LoS wave at a pedestrian, but the proposed method estimates radiation angle of the LoS wave from a vehicle.
Point 5: furthermore, the test conditions should be better defined and differentiated. For example, only LoS is analyzed? what if there is an obstacle in the meantime?
Response 5: Thanks for this comment.
Both the proposed method and the base method detect whether the LoS wave exists and only the LoS wave is used for positioning calculation. We have added Figure 3 to illustrate the difference between LoS signal and Non-LoS signal, and explains how to detect the presence of a LoS wave, in Section 3.5, as follows.
3.5. Presence Detection of the LoS Wave
Presence of the LoS wave is detected by the same method used in the base method [11], which is described in Figure 3. When a LoS wave does not exist, the signal is not used for the position calculation.
We had also added/fixed sentences that shows more detail about the simulation in Section 4.1 and 4.4, as follows.
line 279-280:
In the ray tracing, signal obstruction/reflection by the terrain (ground, building etc.) is simulated.
line 281-283:
After the ray tracing simulation, for each scene, the presence detection of the LoS wave, the estimation of pedestrian-vehicle distance/angle and the position calculation are performed by MATLAB using the acquired signal propagation information.
line 285-287
Satellites with elevation angles no more than 15 degrees and vehicles with distance longer than 40m or without a LoS wave are not used in the calculations, because they are considered to have large ranging errors.
line 327-328:
In order to acquire CSI after the time interval ∆t, additional 3D ray tracing simulation that reflects vehicle/pedestrian movement during ∆t is performed at each scene.
Reviewer 2 Report
The article considers a pedestrian-to-vehicle communication scenario, where a pedestrian device obtains the relative position with respect to the vehicle by overhearing the vehicle-to-vehicle communication signals and estimating the pedestrian position/angle by using channel state information. The paper proposes a method to estimate the signal radiation angle that is based on Doppler information and tracks the phase change of the CSI corresponding to multiple locations of the vehicle.
Overall the paper is sufficiently clear and mathematically sound. The proposed method might be acceptable theoretically, however I believe it needs a more solid validation in a more realistic/representative environment. The authors provided a description of the simulations, however these seem quite simple. For example, in wireless MIMO systems, angle estimation is usually strongly affected by time-varying multipath profiles. Moving vehicles change such profiles in many cases arbitrarly, as they also depend on the surrounding environment (building blockages, reflections from terrain etc...).
Besides, it is unclear how the proposed method for pedestrian localization could outperform GPS based positioning, considering that, as far as I understood, vehicle positioning is obtained from GPS system while pedestrian localization should be affected by the same error bounds.
Author Response
Point 1: I believe it needs a more solid validation in a more realistic/representative environment. The authors provided a description of the simulations, however these seem quite simple. For example, in wireless MIMO systems, angle estimation is usually strongly affected by time-varying multipath profiles. Moving vehicles change such profiles in many cases arbitrarily, as they also depend on the surrounding environment (building blockages, reflections from terrain etc...).
Response 1: Thanks for this comment.
As for building blockage, in the proposed/base method, the presence of a LoS wave is detected, and if a LoS wave does not exist, the vehicle that transmits the signal will not be used for positioning. Therefore, signal blockages simply decrease the number of available vehicles, so we performed the simulation in an environment with few obstacles to simplify the validation. The impact caused by the decrease of the number of available vehicles is evaluated in Section 5.2.2. We have added a section about the presence detection of LoS wave as Section 3.5, as follows.
3.5. Presence Detection of the LoS Wave
Presence of the LoS wave is detected by the same method used in the base method [11], which is described in Figure 3. When a LoS wave does not exist, the signal will not be used for the position calculation.
Different from MIMO systems which try to exploit the scattering of wireless environments, in the proposed method, it is the LoS wave that is used, which will not change much within a short period. As for reflection, this is simulated by the 3D ray tracing, which depends on the instantaneous locations (wireless environment) of the pedestrian and the vehicles.
We have added/revised the following sentences to clarify this, in Section 4.1 and 4.4, as follows.
line 279-280:
In the ray tracing, signal obstruction/reflection by the terrain (ground, building etc.) is simulated.
line 281-283:
After the ray tracing simulation, for each scene, the presence detection of the LoS wave, the estimation of pedestrian-vehicle distance/angle and the position calculation are performed by MATLAB using the acquired signal propagation information.
line 285-287:
Satellites with elevation angles no more than 15 degrees and vehicles with distance longer than 40m or without a LoS wave are not used in the calculations, because they are considered to have large ranging errors.
line 327-328:
In order to acquire CSI after the time interval ∆t, additional 3D ray tracing simulation that reflects vehicle/pedestrian movement during ∆t is performed at each scene.
Point 2: it is unclear how the proposed method for pedestrian localization could outperform GPS based positioning, considering that, as far as I understood, vehicle positioning is obtained from GPS system while pedestrian localization should be affected by the same error bounds.
Response 2: Thanks for this comment.
As pointed out, vehicle positioning uses GPS system, but vehicles also use some additional methods to improve the positioning result. Therefore, the positioning precision of vehicles is much higher than that of pedestrians. We mentioned this in the introduction part, as follows.
This has different impacts on pedestrians and vehicles. Pedestrians mainly rely on satellite positioning, and usually have poor positioning precision in urban canyons. As for vehicles, even though satellite positioning alone cannot provide sufficient precision, its combination with Dead Reckoning, map matching and lane detection etc. helps to keep positioning errors within a few meters [7, 8]. In the future, it is expected that the positioning precision of vehicles will be further improved for autonomous driving, which will be much higher than that of pedestrians.
The performance of GPS positioning for pedestrians is degraded in urban canyons due to the lack of satellites under the shielding of roadside buildings. This is the reason why the proposed method could improve positioning precision of pedestrians by using nearby vehicles as extra anchors. We have fixed sentences about the overview of the base method in the introduction part, as follows.
With the reasonable assumption that vehicle positions have much higher precision that that of a pedestrian, our previous work proposed a method that uses vehicles as anchors in addition to GPS satellites, to improve pedestrian positioning precision [9,10].
Round 2
Reviewer 2 Report
The author improved the manuscript with respect to the quality of the presentation. Considering the reply letter and this new revision, i feel that the technical/scientific soundness of the paper is not sufficient for publication and it must be improved. My recommendation is thus to reject the paper as the requested revision is still major.
The proposed approach for the estimation of Pedestrian-vehicle Angle (the original contribution of the paper, as far as i understood), in particular equations (2)-(8), might work in theory but it is hard to me to believe that this works also in practice.
For example, eq. (2) and eq (3) assume that the relative phase change between the transmitter and the receiver as the vehicle moves only depends on the relative distance delta_d. The above assumption is not acceptable in a real environment as it requires perfect PHY symbol level synchronization between the transmitter and the receiver, that is difficult to achieve in practice. Besides, in the same equations (2) and (3) Doppler effects are also missing.
The proposed approach thus might work in a lab environment (i.e. using a tracking generator and a spectrum analyzer) but it does not work in a real environment. This is also the reason why multiple antennas and standard methods might give some practical benefits in angle estimation. Synthetic aperture radar methods can be also used to (virtually) increase the number antennas although, they require perfect knowledge of vehicle movements.
Author Response
Point 1: eq. (2) and eq (3) assume that the relative phase change between the transmitter and the receiver as the vehicle moves only depends on the relative distance delta_d. The above assumption is not acceptable in a real environment as it requires perfect PHY symbol level synchronization between the transmitter and the receiver, that is difficult to achieve in practice.
Response 1: Thanks for this comment.
In eq.(2) and eq.(3), we are talking about the phase change of the LoS wave, due to the distance variation delta_d (under the movement of vehicles and pedestrian) within the duration delta_t, and the receiver (pedestrian device) measures the phase variation within delta_t. It is not the phase difference between the transmitter and the receiver, but instead the phase difference at two different timings at the receiver side.
Two factors may affect the measurement of phase difference delta_phi within delta_t.
(1) Carrier frequency variation, which may affect initial phase phi_0
(2) Symbol rate variation, which may affect signal receiving.
Although in OFDM system offset in carrier frequency may be large, the symbol rate is relatively stable otherwise the communication fails.
If delta_t is large and CSI is measured in different packets, each packet has a different initial phase offset (phi_0) which changes with the transmitter clock. To avoid this problem, in the proposed method, CSI is detected at different points (OFDM symbols) of the same packet. Because phi_0 is the same, it will be removed in calculating phase different delta_phi.
CSI is measured per OFDM symbol. With the reasonable assumption that OFDM symbol period is accurate, being the same at the transmitter and receiver side, delta_t is measured by counting a pre-defined number of OFDM symbols, at the receiver side.
Therefore, the proposed method doesn’t need time synchronization between the transmitter and the receiver.
The original description might be confusing, and we have modified Section 3.3.2 to make this clear, as follows.
Vehicles move at variable speeds and the time error between vehicles and the pedestrian device also changes, which may affect the phase computation. When acquiring CSI from multiple packets, accurate synchronization of transmission time is required, otherwise, each packet may have a different initial phase. This is difficult. Therefore, the proposed method obtains CSI from multiple locations within a single packet with time interval ∆t, and uses the phase change of the LoS wave derived from the CSI. In OFDM signals used for V2V communication, CSI is calculated from the preamble near the beginning of a packet, and updated per OFDM symbol thereafter by using the pilot signal in subsequent OFDM symbols. Then, strength/phase of the LoS wave is acquired from each CSI (OFDM symbol) and the phase change ∆φ is calculated. Here, ∆t is converted to the number of OFDM symbols, and strength/phase of the LoS wave is acquired from two CSI (two OFDM symbols spaced by ∆t). Then, their phase difference ∆φ is calculated at the pedestrian device, which does not require time synchronization with vehicles.
Point 2: in the same equations (2) and (3) Doppler effects are also missing.
Response 2: Thanks for this comment.
Eq. (2), (3) and (4) explain the phase change of the signal due to Doppler shift by using change of signal propagation distance under vehicle movement. So we consider Doppler shift is already involved in . We have revised the following sentence to clarify this, in Section 3.3.1, as follows.
From equation (2) and (3), , the variation of signal phase of the LoS wave, due to the distance variation , is computed as follows, which corresponds to the Doppler shift.